# Thermoelectric Characteristics of Silicon Carbide and Tungsten-Rhenium-Based Thin-Film Thermocouples Sensor with Protective Coating Layer by RF Magnetron Sputtering

**DOI:** 10.3390/ma12121981

**Published:** 2019-06-20

**Authors:** Zhongkai Zhang, Bian Tian, Yan Liu, Zhe Du, Qijing Lin, Zhuangde Jiang

**Affiliations:** 1State Key Laboratory for Mechanical Manufacturing Systems Engineering, Xi’an Jiaotong University, Xi’an 710049, China; z.zhongkai@stu.xjtu.edu.cn (Z.Z.); liuyanzj402@stu.xjtu.edu.cn (Y.L.); duzhechn@stu.xjtu.edu.cn (Z.D.); linqijingcor@163.com (Q.L.); zhuangdejiangcor@163.com (Z.J.); 2Institute of Materials in Electrical Engineering 1, RWTH Aachen University, 52074 Aachen, Germany

**Keywords:** Film, sensor, thermoelectricity

## Abstract

A thin-film thermocouples (TFTCs) sensor based on silicon carbide substrate, 95 wt% tungsten–5 wt% rhenium (W-5Re) and 74 wt% tungsten–26 wt% rhenium (W-26Re) thermosensitive part with aluminum oxide protective coating layer was designed and fabricated by radio frequency (RF) magnetron sputtering. It exhibited a high thermoelectric voltage of 35.51 mV when the temperature difference was 1240 °C (the hot junction temperature was 1420 °C), with an average Seebeck coefficient of 28.63 µV/°C, which was 27% larger than the standard C-type thermocouple wires at the same temperature difference. The repeatability error was ±4.1%, the drift rate was 9.6 °C/h for 10 h and the laser response time was 0.36 ms. Compared to the traditional thermocouple, it could provide long-term temperature testing within 1420 °C for the requirement of high-temperature measurement and high response speed.

## 1. Introduction

High-temperature measurement in a narrow space, such as a combustion chamber in an engine, imposes harsh requirements for thermosensitive parts to meet [1,2,3,4,5]. Thin-film thermocouples (TFTCs) are suitable for in situ and real-time measurement with no airflow disturbance of the environment for its nanoscale thickness and negligible mass [6,7,8,9]. TFTCs are widely used as a new type of thermal sensor [10,11,12]. In the 1980s, the National Aeronautics and Space Administration (NASA) Lawrence Center had already started research into TFTCs for turbine blade surface temperature measurements and successfully deposited nickel-chromium/nickel-aluminum (Ni–Cr/Ni-Al) thermocouples on a metal-based blade [13]. Platinum and palladium TFTCs are extremely easily oxidized [14,15]. Ceramic materials have been used in TFTCs for their stronger oxidation resistance and stable chemical structure. The Virginia Polytechnic Institute and State University had presented tantalum carbide/titanium carbide (Ta–C/Ti–C) TFTCs by radio frequency magnetron sputtering, the film failure occurred at temperatures above 1070 °C [16]. Indium tin oxide (ITO) has a faster volatilization speed at temperatures over 1100 °C and at melting points below 1200 °C [17]. The TFTCs reported recently can hardly work for a long time in an air environment with a temperature higher than 1400 °C. Tungsten-rhenium TFTCs show considerable promise as TFTC temperature sensors because of their high melting point (at about 2800 °C). However, the oxidation of tungsten-rhenium TFTCs results in Seebeck coefficient changes with increasing service temperature and time, which lead to the change of thermoelectric characteristics on tungsten-rhenium TFTC sensors, even resulting in failure. To measure the high temperature with low airflow interference in the engine, like F119 type fighter (about 1670 °C), it is important to discuss the thermoelectric characteristics of tungsten-rhenium TFTCs at high temperatures and present a way to avoid the failure caused by oxidation.

A tungsten-rhenium TFTC sensor was reported in the *Review of Scientific Instruments* [18] by our group. In this paper, we optimized the structure, process, and added a protective coating layer to increase the service temperature to 1420 °C. Firstly, the sensor was designed by the simulation results. Secondly, 95 wt% tungsten–5 wt% rhenium (W-5Re) and 74 wt% tungsten–26 wt% rhenium (W-26Re) TFTCs were fabricated by radio frequency (RF) magnetron sputtering. The TFTCs with a protective coating layer, which mainly contained aluminum oxide (Al_2_O_3_), were prepared and compared with the uncoated sample. Finally, the repeatability error, electromotive force (EMF) behavior and the constant time of the carbon dioxide (CO_2_) laser pulse of the TFTCs with a protective coating layer were investigated.

## 2. Materials and Methods 

Equation (1) indicates the relationship between the output voltage and temperature, where *V* is the voltage output of thermocouples, *S_A_* and *S_B_* represent the Seebeck coefficient of two kinds of materials. *T*_1_ and *T*_2_ represent the temperature of the cold and hot junction.(1)$$V=\int_{T_{1}}^{T_{2}}\left(S_{A}-S_{B}\right)Tdt$$

The Seebeck coefficient *S* is influenced by basic physical quantities, such as the Fermi level, effective mass, relaxation time and scattering mechanism, which can be described as Equations (2) and (3) in the metal thermocouple material, when no temperature gradient exists:(2)$$S=\mp \frac{1}{eT}\left[\frac{K_{1}}{K_{2}}-E_{F}\right]$$(3)$$K_{i}=\int_{0}^{\infty }\tau U_{x}^{2}g\left(E\right)E^{i-1}\frac{\partial f_{0}}{\partial E}dE,\left(i=1\sim 3\right)$$where $f_{0}$ refers to the distribution function of the carrier in the equilibrium state, h refers to the characteristic parameters in the relaxation process, $E_{F}$ refers to the Fermi level, τ refers to the dielectric constant, e refers to the electric charge of a single electron. Equation (4) indicates the density of the states of a carrier near the conduction band bottom:(4)$$g\left(E\right)=\frac{4\pi {\left(2m^{*}\right)}^{3/2}}{h^{3}}E^{1/2}$$where m* refers to the effective mass of the carrier, h refers to the Planck constant, and E refers to the energy of the carrier. The carrier is a kind of fermion, which has spin. It obeys the Pauli exclusion principle and Fermi-Dirac distribution.

In general, thermocouples are classified into two types: Metal type and semiconductor type, all based on the Seebeck effect, but the sensing mechanism is slightly different. Common semiconductor types such as ITO/In_2_O_3_, La_X_Sr_1−x_CrO_3_ (LSCO)/ITO, will form a p-n type junction. The Seebeck coefficient for both materials needs to be considered individually. However, the metal type, as discussed here, does not form a p-n type junction. The metal type works in pairs for both materials to make different free electron diffusion rates and a single Seebeck coefficient for each part, like W-5Re and W-26Re separately, and is not usually characterized.

Thin films were sputtered by RF magnetron sputtering (DISCOVERY, 635, Moorestown, NJ, US) from high purity 101.6 mm diameter W-5Re and W-26Re targets (purity 99.99%, ZHONGNUO Co., Beijing, China) on a silicon carbide (SiC) ceramic substrate. The structure and process are shown in Figure 1. The SiC substrates were cleaned by ultrasonic with acetone and ethyl alcohol before preparation. The films were patterned using conventional photolithography techniques. Then, the pattern was formed through a mask by UV light exposure (WJH, ABM6, San Jose, CA, US). The hot junction, which determined the thermal load, was made by the superposition film of the W-5Re film and the W-26Re film, while the voltage was outputted from the end of the negative film [19]. The wires were attached at the cold junction by the ceramic jig. We used copper (Cu) foil to make the thermocouple cold junction electrical contacts. Argon (Ar) gas was added to the sputtering after vacuum pumping. The sputtering parameters of the W-5Re and W-26Re thin films are shown in Table 1. The TFTCs were heat-treated at 300 °C for 6 h [20] in the experiment.

A muffle furnace (Nabertherm, P300 LHT 2-17, Lilienthal, Germany) and a carbon dioxide laser generator (AL 30, ACCESSLASER, Shenzhen, China) were used to analyze the static and dynamic thermal response of the TFTCs, as shown in Figure 2. The hot junction was put in the muffle furnace. The cold junction was outside of the furnace. The S-type and K-type standard armored thermocouples (OMEGA, Norwalk, CT, US) were attached on the hot and cold junction separately to measure the temperature. The hot and cold junction temperature and the TFTC output voltage data were recorded using a data recorder (HIOKI, LR8431-30, Nagano-ke, Japan). Laser response experiments were done. A function generator (Tektronix, 2200, Shanghai, China) was used to help provide a dynamic thermal shock of the laser generator. The signal was collected by a high-frequency oscilloscope (Tektronix, MDO3012, Shanghai, China). In order to analyze film changes, the component of the TFTCs was taken by X-ray diffraction (XRD, Rigaku Corporation, d/max-2400, Beijing, China).

## 3. Results and Discussion

The stress finite element model was set to design the size parameters of the TFTCs [21]. In the thermal stress finite element analysis (FEA) by ANSYS (ANSYS Inc., Berkeley, CA, US), the isotropic, thermoplastic and orthotropic behavior of the material was considered. The relationship between the thermal stress and thermoelectric characteristics was analyzed, as shown in Figure 3. The thermal stress between the W-5Re and W-26Re film ranged from 74 MPa at 400 °C to 270 MPa at 1400 °C, which was compressive stress. The voltage of the TFTCs ranged from 12 mV at 400 °C to 37 mV at 1200 °C in the temperature difference. It provided detectable electrical signals with thermal stress less than the fracture limit. The substrate thickness was 2 mm, the line width of the TFTCs was 2 mm, the thickness of the W-5Re and W-26Re film were 2 µm, and the length of positive and negative parts was 8 cm.

In order to analyze film changes, the sections and surface of the TFTCs were taken with a scanning electron microscope (SEM, type: Tescan Mira 3, Shanghai, China), as shown in Figure 4.

The post-processing instrument was designed to compensate for the cold junction temperature and record data, which is shown in Figure 5c,d. The relationship between the Seebeck coefficient and temperature was set at the system of the post-processing instrument to realize data conversion. Considering the Seebeck coefficient changed slowly with the temperature, the mapping table of the output voltage and temperature was set in each to one degree. The post-processing instrument could be divided into two parts: The digital part and the analog part. The analog part complete the output voltage of the thermocouple acquisition and conditioning, including the analog to digital conversion circuit (ADC) and ADC drive circuit, and the chip power supply; the digital part read the ADC conversion results at the same time to get the cold injunction temperature through the temperature sensor. The temperature data was displayed. In addition, the digital part had local storage and a universal serial bus (USB) interface, which was convenient for the communication of the host computer. The TFTCs coated with the aluminum oxide (Al_2_O_3_) protective coating layer were prepared by RF sputtering to compare with the uncoated sample. Figure 5 shows the TFTC sample, TFTC sample with the coating layer, post-processing instrument, and post-processing logic diagram.

The XRD patterns of the W-5Re and W-26Re film serviced at 0 °C, 600 °C and 1420 °C in air are shown in Figure 6a,b separately to analyze the influence of oxidation on the tungsten-rhenium TFTCs. It could be observed that the film had a strong (200) preferred orientation. As the temperature increased to 600 °C for a long time, the intensity of the diffraction peak changed. The tungsten was oxidized into tungsten trioxide (WO_3_) when compared with the XRD standard card (JADE card, No.72-1465), shown in Figure 6c. It could be observed from Figure 6b that the situation was approximately the same for W-26Re. The component of the films was stable at 1420 °C after the coating layer (Al_2_O_3_) was added. The peak of the film without a coating layer at 1420 °C still demonstrated oxidation of tungsten, similar to the peak at 600 °C. In this experiment, the thickness of the tungsten-rhenium film in the sample was 2 µm and the thickness of the Al_2_O_3_ coating layer was 20 nm. Since the coating layer was very thin and with regard to transmissivity, tungsten-rhenium is a metal with high reflectivity, it was difficult to observe the peak of the coating layer on it. In addition, here we focused on the oxidation of W-5Re and W-26Re, the peak of Al_2_O_3_ was not an important point of observation for its high thermal stability. The SEM (Tescan Mira 3, Shanghai, China) section image of W-5Re and W-26Re with a coating layer is shown in Figure 6d. The coating layer was dense, which could help to avoid oxidation of the TFTCs. The porosity of the coating layer may lead the uncoated part of the film to become oxidized and determines the stability of the TFTCs. It was hard to control the density, which requires adjustment of the preparation process for the coating layer. This, however, is not the focal point of this article. A four-probe conductivity tester (ST2258C, Suzhou Jingge Electronic Co., Suzhou, China) was used to test the electrical conductivity changes during the service. The conductivity of W-5Re changed from 10.83 S/m at room temperature to 0.018 S/m at 600 °C after four hours. The conductivity of W-5Re at 1420 °C with a coating layer and without a coating layer was 8.62 S/m and 0.003 S/m, respectively. The conductivity was measured at room temperature after heating, which was limited by the service temperature of the four-probe conductivity tester. The electrical conductivity reduced by nearly 1000 times in the oxidation process, which obeyed the oxidation phenomena in the XRD results. The oxidation temperature of rhenium was relatively high. It began at 1500 °C and the relative content was small in this case. It was a secondary factor in this. This paper focuses on the oxidation of tungsten. It can be observed that the thermosensitive part composed of W-5Re and W-26Re film was oxidized at a high temperature and a coating layer process could avoid the TFTCs from being oxidized.

The oxidation and compositional changes of the tungsten-rhenium TFTC ultimately affect the thermoelectric properties, including the Seebeck coefficient and repeatability error, and lead to the failure of TFTCs, which is shown in Figure 7a. A coating layer process can not only avoid the TFTCs from being oxidized but ensure its Seebeck effect to yield a thermoelectric voltage at 1420 °C, which is shown in Figure 7b.

In Figure 7a, a series of output voltages were collected for the heating process of the tungsten-rhenium TFTCs. The two curves had a similar linear shape, while the tungsten-rhenium with the coating layer had a better linear relationship with temperature difference. In addition, for the uncoated TFTC, it had a higher output voltage than that of the coated sample. This was due to the reduction of the hot junction temperature caused by the heat insulation effect of the coating layer. When the tungsten-rhenium TFTCs suffered from a higher temperature, up to nearly 600 °C for a long time (Figure 7a), the uncoated TFTCs failed after a long time of service because of oxidation. Therefore, the coating layer could enhance the high-temperature performance of the tungsten-rhenium TFTCs prominently. The repeatability of the TFTCs was tested in three cycles, as shown in Figure 7b. The TFTCs tracked with the hot junction temperature very well. There were no failures of the TFTCs up to 1420 °C. Therefore, tungsten-rhenium TFTCs with an Al_2_O_3_ coating layer are promising thermal sensors for high-temperature sensing.

The maximum standard deviation of the three cycles was 1.048 mV. The repeatability error of the coated tungsten-rhenium TFTCs was ±4.1%. The peak thermoelectric voltage of the TFTCs with protection was 35.51 mV when the temperature difference between the hot junction and cold junction was 1240 °C (the hot junction temperature was 1420 °C). The average Seebeck coefficient at 1240 °C reached 28.63 µV/°C, 27% larger than the average Seebeck coefficient of the standard C-type (tungsten-rhenium, 22.51 µV/°C at 1240 °C [22]) thermocouple at 1240 °C for the component difference in RF magnetron sputtering between the metal wire and film. The Seebeck effect is influenced by basic physical quantities, such as the Fermi level, effective mass, relaxation time, scattering mechanism and carrier density. The standard C-type was made of tungsten-rhenium alloy wire. These physical quantities may be changed by magnetron sputtering of tungsten-rhenium compared with the wire due to the size effect [23].

Equation (5) is used to describe the EMF behavior:*E*(*T**) = *A* × (*T**)^2^ + *B* × (*T**) + *C*(5)

Here *T** refers to the temperature difference between the hot junction and cold junction; the unit of *T** is °C. V refers to the output voltage; the unit of E is mV. The *R*^2^ values for the fittings were 0.9992. The parameter C was arbitrarily set to zero as the boundary condition. *T** = 0 must be satisfied in the EMF behavior formula to facilitate the practical application. The results are shown in Table 2. In addition, the constant time of the coated TFTCs and uncoated TFTCs was tested. We measured a 95% response time of about 0.9 ms and 0.36 ms, respectively. This showed the influence of the coating layer on heat transfer and the response was very small. Compared with the response time of the armored thermocouple wire (about 1 s), the tungsten-rhenium TFTCs showed a great application prospect to meet the requirements of fast response measurements.

The thickness of the Al_2_O_3_ coating layer here was 500 nm. We tested the coating layers with different thicknesses to optimize the thermoelectric response. The Seebeck coefficient was 28.63 µV/°C, 28.56 µV/°C, 28.53 µV/°C when the thickness of the Al_2_O_3_ coating layer was 500 nm, 1 µm and 2 µm, respectively. It could be observed that the thermoelectric characteristics did not change with the thickness of the Al_2_O_3_ coating layer. The thermoelectric characteristics were determined by tungsten-rhenium, not by the influence of the protective layer due to the Seebeck effect.

The TFTCs exhibited repeatability at the test temperature. In the third heat cycle, the maximum output voltage of the thermocouple dropped by 0.3 mV, which may be due to additional annealing effects and volatilization in the heating cycles. The drift rate ε was set to indicate the stable service time of the TFTCs, as shown in Equation (6):(6)$$\varepsilon \left(T\right)=\frac{△V\left(T\right)}{Vref\left(T\right)}\cdot \frac{T}{△t}$$

Here △V(T) is the voltage drop, Vref(T) is the initial thermoelectric voltage and △t is the lifetime during which the thermocouple soaked at a particular temperature T for the hot junction. The optimized TFTCs were heated at 1420 °C for 10 h to test the drift rate. The drift rate was 9.6 °C/h. Compared with the tungsten-rhenium TFTCs we presented before, the highest stability service temperature increased to 1420 °C and the stable service time increased from 2 h to 10 h. Compared with the current TFTCs, such as ITO 90/10 vs. In_2_O_3_, which has a peak voltage at 1200 °C for the hot junction temperature and a drift rate of 24.06 °C/h [24], or the traditional Type-K armored thermocouple in air [25], this tungsten-rhenium TFTCs sensor provided better performance during in situ and real-time temperature measurements at 1420 °C for a long time.

## 4. Conclusions

A silicon carbide ceramics-based tungsten-rhenium TFTC sensor was prepared by RF sputtering. The TFTCs coated with an Al_2_O_3_ layer were prepared and compared with the uncoated sample. The experiment showed the coating layer avoided the failure of the TFTCs, caused by oxidation at nearly 600 °C effectively and increased the stable maximum service temperature to 1420 °C when the cold junction temperature was kept at 0 °C. The average Seebeck coefficient at 1240 °C reached 28.63 µV/°C, which was 27% larger than the average Seebeck coefficient of the standard C-type. The repeatability error was ±4.1% and the drift rate was 9.6 °C/h for 10 h. This solved the problem that conventional thin film thermocouples cannot directly work at a high temperature of 1420 °C for a long time. The coated tungsten-rhenium TFTCs provide application value for high-temperature sensing in air.

## Figures and Tables

**Figure 1 materials-12-01981-f001:**
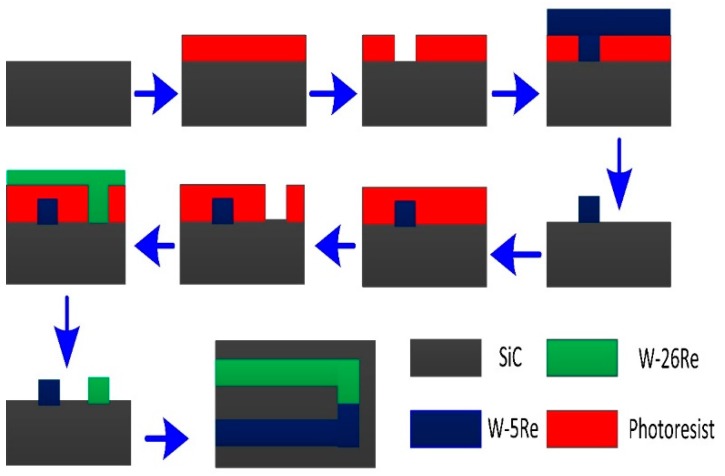
Brief schematic of tungsten-rhenium thin-film thermocouples (TFTCs).

**Figure 2 materials-12-01981-f002:**
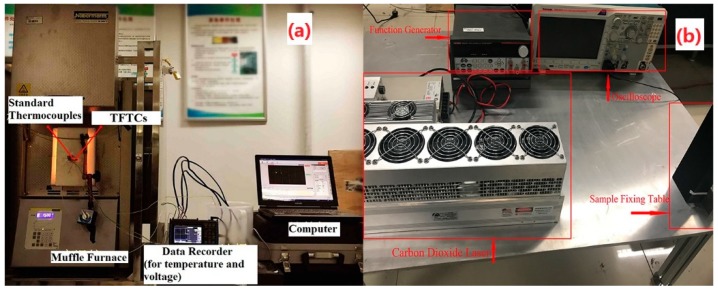
Static (**a**) and dynamic (**b**) high-temperature thermoelectric measurement experiment rig.

**Figure 3 materials-12-01981-f003:**
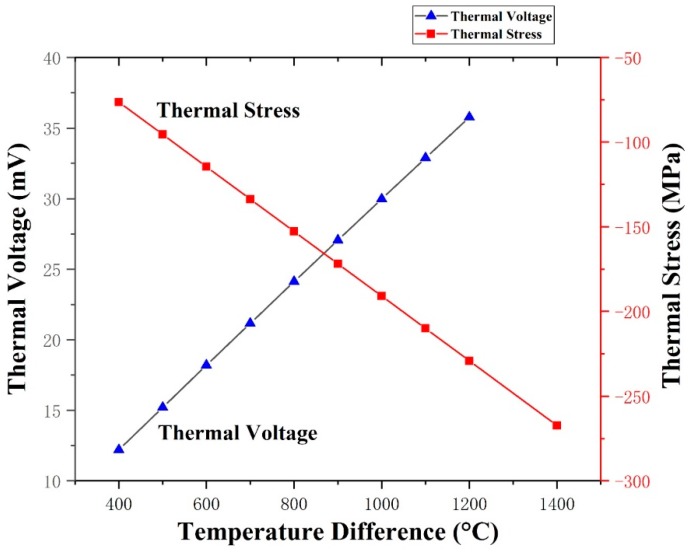
The relationship between thermal stress and thermoelectric characteristics by finite element analysis (FEA).

**Figure 4 materials-12-01981-f004:**
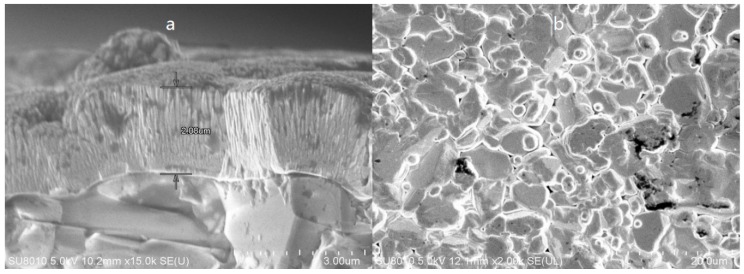
SEM images of the 95 wt% tungsten–5 wt% rhenium (W-5Re) section (**a**) and surface (**b**) after heat treatment.

**Figure 5 materials-12-01981-f005:**
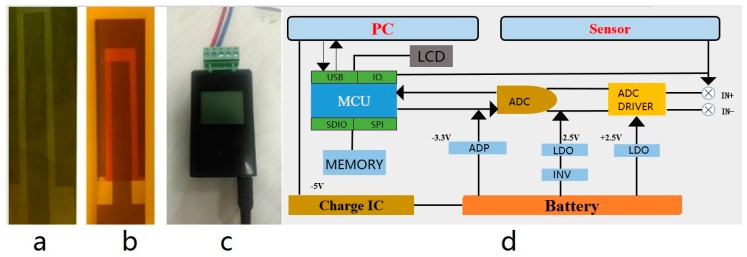
TFTCs sample (**a**), TFTCs sample with coating layer (**b**), post-processing instrument (**c**), post-processing logic diagram (**d**).

**Figure 6 materials-12-01981-f006:**
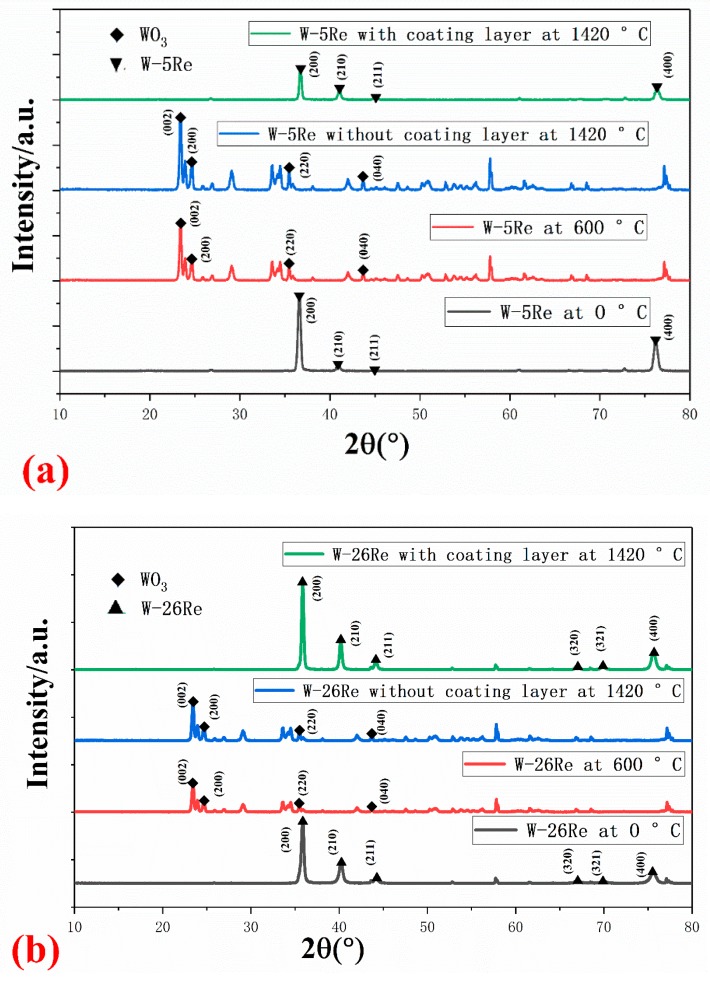
XRD patterns of W-5Re (**a**) and W-26Re thin-film and coated thin-film at different temperatures (**b**), XRD standard card (**c**) of WO_3_, and SEM section image (**d**) of W-5Re with coating layer.

**Figure 7 materials-12-01981-f007:**
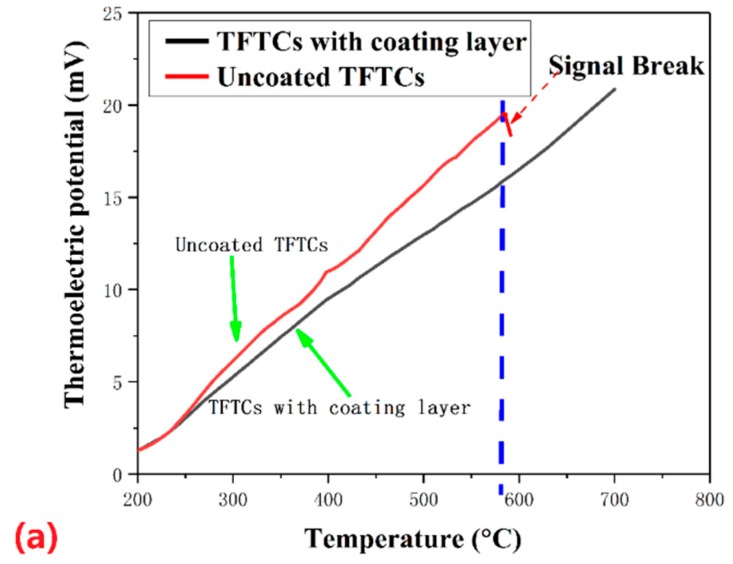
Thermoelectric output of tungsten-rhenium TFTCs in heating at 700 °C (**a**), and in cycle of heating and cooling with a peak hot junction temperature of 1420 °C (**b**).

**Table 1 materials-12-01981-t001:** Sputtering parameters of 95 wt% tungsten–5 wt% rhenium (W-5Re) and 74 wt% tungsten–26 wt% rhenium (W-26Re) thin films.

Sputtering Parameters	Targets	
	W-5Re	W-26Re
**Target Diameter (mm)**	101.6	101.6
**Argon Flow Rate (Sccm)**	30	60
**Sputtering Power (W)**	400	200
**Vacuum Degree (Torr)**	0.7 × 10^−6^	1 × 10^−6^

**Table 2 materials-12-01981-t002:** Polynomials coefficients of the TFTCs.

A(mV/°C^2^)	B(mV/°C)	C(mV)	Average Seebeck Coefficient at 1240 °C (µV/°C)	Average Seebeck Coefficient of Standard C-type Thermocouple 1240 °C (µV/°C)
−0.8627 × 10^−6^	0.03084	0	28.63	22.51

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
