# Peer review of "Thermoelectric Characteristics of Silicon Carbide and Tungsten-Rhenium-Based Thin-Film Thermocouples Sensor with Protective Coating Layer by RF Magnetron Sputtering"

_materials, 2019, doi:10.3390/ma12121981_

Round 1
Reviewer 1 Report
This paper reports the thin film thermocouples sensor based on tungsten and rhenium with the aim to sustain the harsh high temperature environments, such the narrow space.
The idea is interesting, but the manuscript includes several points to be corrected, as follows.
1) The structure of the paper is confusing. The figures 1, 3 and 4 should appear when the authors are discussing the results, and not in the section 2. Materials and Methods. In particular, the figure 3 (SEM results) need to be discussed in the section 3, and therefore it should not be appear in the materials and methods section.
2) Please include the thickness of Al2O3 coating layer. The authors tested coating layers with different thicknesses to optimize the thermoelectric response?
3) In the figure 6, please specify the XRD peaks which corresponds to WO3, and the code number XRD standard cards for the elements. In the first paragraph of section 3, the authors compare the XRD data with thermoelectric performance but no evidence on Seebeck effect is here analysed.
4) Authors refers that “the XRD situation is approximately the same for W-26Re”. Please include the XRD data and analysis for this element.
5) Please include the electrical conductivity value of both films at 1420ºC.
6) Please include in the caption of figure 7, figure (a) and (b).
7) Which is the material of electrical contacts at the cold side of the thermocouple?
8) The authors refer that the obtained experimental Seebeck voltage is 27% larger than the standard value of C-type (tungsten-rhenium). Please include the reference. The explanation for the difference is very widespread “This change may be caused by the magnetron sputtering has changed the basic physical quantities of tungsten-rhenium TFTCs”. Please provide more discussion based on literature references.
9) The thermocouple is based on a p-n junction, n or p-type? This will be referred in the paper.
10) Seebeck response should be provided for both materials individually, in order to compare the average seebeck coefficient of the device to the individual elements.
11) Some sentences are not very perceptible in the manuscript (such as, “This change may be caused by the magnetron sputtering has changed the basic physical quantities of tungsten-rhenium TFTCs”). Please review all the manuscript and clarified the sentences.
Author Response
Response to Reviewer 1 Comments
Dear Reviewers:
On behalf of my co-authors, we thank you very much for giving us an opportunity to revise our manuscript, we appreciate editor and reviewers very much for their positive and constructive comments and suggestions on our manuscript entitled " Thermoelectric characteristics of Silicon Carbide and Tungsten-Rhenium Based Thin-Film Thermocouples Sensor with Protective Coating Layer by RF Magnetron Sputtering" (materials-512151). We have studied reviewers’ comments carefully and have tried our best to revise our manuscript according to the comments.
We did some corrections according to your suggestion, some are mentioned here:
Point 1: The structure of the paper is confusing. The figures 1, 3 and 4 should appear when the authors are discussing the results, and not in the section 2. Materials and Methods. In particular, the figure 3 (SEM results) need to be discussed in the section 3, and therefore it should not be appear in the materials and methods section.
Response 1: The structure of the paper is revised. The figures 1, 3 and 4 were changed to be discussed in the section 3 (new fig.3,4,5), and the related paragraph were also revised.
Point 2: Please include the thickness of Al2O3 coating layer. The authors tested coating layers with different thicknesses to optimize the thermoelectric response?
Response 2: the thickness of Al2O3 coating layer was added. We had tested coating layers with different thicknesses to optimize the thermoelectric response, the Seebeck coefficient doesn't changes. Some results were added below table.2.
Point 3: In the figure 6, please specify the XRD peaks which corresponds to WO3, and the code number XRD standard cards for the elements. In the first paragraph of section 3, the authors compare the XRD data with thermoelectric performance but no evidence on Seebeck effect is here analysed.
Response 3: figure 6 was revised, the XRD peaks were noted and the code number XRD standard cards were added. The Seebeck effect was analysed after compare the XRD data.
Point 4: Authors refers that “the XRD situation is approximately the same for W-26Re”. Please include the XRD data and analysis for this element
Response 4: The analysis of XRD data and W-26Re was added below Figure 6. In addition, the XRD pattern of W-26Re were not added due to the extremely small difference between W-5Re.
Point 5 Please include the electrical conductivity value of both films at 1420ºC.
Response 5: The electrical conductivity value of both films at 1420ºC was mentioned in the first paragraph of part.3.
Point 6: Please include in the caption of figure 7, figure (a) and (b).
Response 6: The caption of figure 7 has been revised.
Point 7: Which is the material of electrical contacts at the cold side of the thermocouple?
Response 7: We use copper foil to make thermocouple cold junction electrical contacts. I add a description above Table 1.
Point 8: The authors refer that the obtained experimental Seebeck voltage is 27% larger than the standard value of C-type (tungsten-rhenium). Please include the reference. The explanation for the difference is very widespread “This change may be caused by the magnetron sputtering has changed the basic physical quantities of tungsten-rhenium TFTCs”. Please provide more discussion based on literature references.
Response 8: The reference and more discussion were added above part.4. The main reason is the basic physical quantities are changed by magnetron sputtering of tungsten-rhenium.
Point 9 & 10: The thermocouple is based on a p-n junction, n or p-type? This will be referred in the paper.
Seebeck response should be provided for both materials individually, in order to compare the average seebeck coefficient of the device to the individual elements.
Response 9 & 10: I added a paragraph at part.2 to explain it. In general, thermocouples are classified into two types: metal type and semiconductor type, all based on the Seebeck effect, but the sensing mechanism is little different. Common semiconductor types such as ITO/In2O3, LaXSr1-xCrO3 (LSCO)/ITO, will form a p-n type junction. Seebeck coefficient for both materials individually need to be considered. However, the metal type, as what we discussed here, does not form a p-n type junction. The metal type work in pairs for both materials to make different free electron diffusion rates, and a single Seebeck coefficient for each part, like W-5Re and W-26Re separately, is not characterized usually.
Point 11: Some sentences are not very perceptible in the manuscript (such as, “This change may be caused by the magnetron sputtering has changed the basic physical quantities of tungsten-rhenium TFTCs”). Please review all the manuscript and clarified the sentences.
Response 11: Some words and sentences were revised to make it clearly, and help to have a native English-speaking colleague read the manuscript.

Reviewer 2 Report
The present paper describes an interesting approach to realize high temperature thermocouples in film technology, which are stable against oxidation and provide a still fast response. The material and sample preparation procedure is well described and the measured results are explained in detail.
Some questions are still open. The reviewer recommends a major revision. The following points should be considered before publication.
1. The quality of the figures should be improved: It is not clear in a black & white representation which curves belong to which axes (Fig. 1) or to which setup of the thermocouple (Fig. 6) or to which parameter (Fig. 7). It would get more clear if the legend would be placed directly to each curve.
2. Fig. 6 shows the XRD data of three different thermocouple setups. I miss the comparison to the standard cards of tungsten and tungsten oxide, the XRD patterns of the standards should be added.
3. For better comparison of the coating process, an XRD result of a “W-5Re thermocouple without coating layer at 1420 °C” should be added. Are Al2O3-patterns visible at the XRD pattern of “W-5Re with coating layer at 1420 °C”? Please comment on this?
4. What happens to the rhenium phase at different temperatures? Does rhenium oxidize at high temperatures to Rhenium oxides, too? The authors should comment on this point.
5. It would be interesting to add an SEM image of the coated W-5Re thermocouple (section). This would improve the discussed results with respect to the necessary density of the coating layer. I would expect that the cover layer could only avoid the oxidation of TFTC if it is dense and that the porosity of the Al2O3 layer determines the stability of the TFTC. The discussion of this point is missing in the paper and should be included.
6. On page 6, line 161-165 is written that the Seebeck coefficient is different between standard C-type and the coated TFTC and a short explanation is given. This point should be discussed more in detail, since the Seebeck coefficient is a material property independent on the geometry. Which basic physical quantities are changed by magnetron sputtering of tungsten-rhenium? How can the difference in the Seebeck coefficient of around 27% be explained?
Author Response
Response to Reviewer 2 Comments
Dear Reviewers:
On behalf of my co-authors, we thank you very much for giving us an opportunity to revise our manuscript, we appreciate editor and reviewers very much for their positive and constructive comments and suggestions on our manuscript entitled " Thermoelectric characteristics of Silicon Carbide and Tungsten-Rhenium Based Thin-Film Thermocouples Sensor with Protective Coating Layer by RF Magnetron Sputtering" (materials-512151). We have studied reviewers’ comments carefully and have tried our best to revise our manuscript according to the comments.
We did some corrections according to your suggestion, some are mentioned here:
Point 1 The quality of the figures should be improved: It is not clear in a black & white representation which curves belong to which axes (Fig. 1) or to which setup of the thermocouple (Fig. 6) or to which parameter (Fig. 7). It would get clearer if the legend would be placed directly to each curve.
Response 1: The figures (new Fig.3, Fig.6, Fig.7) were revised to make it clear in both black and white and colour versions.
Point 2: Fig. 6 shows the XRD data of three different thermocouple setups. I miss the comparison to the standard cards of tungsten and tungsten oxide, the XRD patterns of the standards should be added.
Response 2: The XRD peaks were noted, the XRD patterns of the standards and the code number XRD standard cards were added above Fig. 6.
Point 3. For better comparison of the coating process, an XRD result of a “W-5Re thermocouple without coating layer at 1420 °C” should be added. Are Al2O3-patterns visible at the XRD pattern of “W-5Re with coating layer at 1420 °C”? Please comment on this?
Response 3: The XRD result of a W-5Re thermocouple without coating layer at 1420 °C was added in Fig.6. Al2O3-patterns are not visible at the XRD pattern of "W-5Re with coating layer at 1420 °C".. In this experiment, the thickness of the tungsten-rhenium film in the sample was 2 µm, and the thickness of Al2O3 coating layer was 20 nm. Since the coating layer is very thin and transmissivity, tungsten-rhenium is metal which has high reflectivity, it is difficult to observe the peak of coating layer on it. In addition, here we focuses on the oxidation of W-5Re and W-26Re, the peak of Al2O3 is not an important point of observation for its high thermal stability. I have added explains about it above Fig.6.
Point 4: What happens to the rhenium phase at different temperatures? Does rhenium oxidize at high temperatures to Rhenium oxides, too? The authors should comment on this point.
Response 4: Comment about the rhenium was added below Fig.6. The oxidation temperature of rhenium is relatively high, generally begin from 1500 degrees, and the relative content is small in this case. It is a secondary factor in this. This paper focuses on the oxidation of tungsten
Point 5: It would be interesting to add an SEM image of the coated W-5Re thermocouple (section). This would improve the discussed results with respect to the necessary density of the coating layer. I would expect that the cover layer could only avoid the oxidation of TFTC if it is dense and that the porosity of the Al2O3 layer determines the stability of the TFTC. The discussion of this point is missing in the paper and should be included
Response 5: The SEM section image was added and discussed in part.3.
Point 6: On page 6, line 161-165 is written that the Seebeck coefficient is different between standard C-type and the coated TFTC and a short explanation is given. This point should be discussed more in detail, since the Seebeck coefficient is a material property independent on the geometry. Which basic physical quantities are changed by magnetron sputtering of tungsten-rhenium? How can the difference in the Seebeck coefficient of around 27% be explained?
Response 6: The reference and more discussion were added below Fig.7. The Seebeck effect is influence by the basic physical quantities, such as Fermi level, Effective Mass, relaxation time, scattering mechanism and carrier density. The standard C-type was made of tungsten-rhenium alloy wire. These may be changed by magnetron sputtering of tungsten-rhenium compared with the wire be due to size effect.

Round 2
Reviewer 1 Report
Thank you for the corrections in the manuscript. The manuscript is already more structured and therefore, less confusing.
The paper can be accepted in present form.